# A Complete Recipe for Stochastic Gradient MCMC

**Yi-An Ma**, **Tianqi Chen**, and **Emily B. Fox**

University of Washington {`yianma@u,tqchen@cs,ebfox@stat`}`.washington.edu`

## Abstract

Many recent Markov chain Monte Carlo (MCMC) samplers leverage continuous dynamics to define a transition kernel that efficiently explores a target distribution. In tandem, a focus has been on devising scalable variants that subsample the data and use stochastic gradients in place of full-data gradients in the dynamic simulations. However, such stochastic gradient MCMC samplers have lagged behind their full-data counterparts in terms of the complexity of dynamics considered since proving convergence in the presence of the stochastic gradient noise is nontrivial. Even with simple dynamics, significant physical intuition is often required to modify the dynamical system to account for the stochastic gradient noise. In this paper, we provide a general recipe for constructing MCMC samplers—including stochastic gradient versions—based on continuous Markov processes specified via a two matrices. We constructively prove that the framework is *complete*. That is, any continuous Markov process that provides samples from the target distribution can be written in our framework. We show how previous continuous-dynamic samplers can be trivially "reinvented" in our framework, avoiding the complicated sampler-specific proofs. We likewise use our recipe to straightforwardly propose a new state-adaptive sampler: *stochastic gradient Riemann Hamiltonian Monte Carlo* (SGRHMC). Our experiments on simulated data and a streaming Wikipedia analysis demonstrate that the proposed SGRHMC sampler inherits the benefits of Riemann HMC, with the scalability of stochastic gradient methods.

## 1  Introduction

Markov chain Monte Carlo (MCMC) has become a defacto tool for Bayesian posterior inference. However, these methods notoriously mix slowly in complex, high-dimensional models and scale poorly to large datasets. The past decades have seen a rise in MCMC methods that provide more efficient exploration of the posterior, such as Hamiltonian Monte Carlo (HMC) [8, 12] and its Reimann manifold variant [10]. This class of samplers is based on defining a *potential energy* function in terms of the target posterior distribution and then devising various continuous dynamics to explore the energy landscape, enabling proposals of distant states. The gain in efficiency of exploration often comes at the cost of a significant computational burden in large datasets.

Recently, stochastic gradient variants of such continuous-dynamic samplers have proven quite useful in scaling the methods to large datasets [17, 1, 6, 2, 7]. At each iteration, these samplers use data subsamples—or *minibatches*—rather than the full dataset. Stochastic gradient Langevin dynamics (SGLD) [17] innovated in this area by connecting stochastic optimization with a first-order Langevin dynamic MCMC technique, showing that adding the "right amount" of noise to stochastic gradient ascent iterates leads to samples from the target posterior as the step size is annealed. Stochastic gradient Hamiltonian Monte Carlo (SGHMC) [6] builds on this idea, but importantly incorporates the efficient exploration provided by the HMC momentum term. A key insight in that paper was that the naïve stochastic gradient variant of HMC actually leads to an incorrect stationary distribution (also see [4]); instead a modification to the dynamics underlying HMC is needed to account for

the stochastic gradient noise. Variants of both SGLD and SGHMC with further modifications to improve efficiency have also recently been proposed [1, 13, 7].

In the plethora of past MCMC methods that explicitly leverage continuous dynamics—including HMC, Riemann manifold HMC, and the stochastic gradient methods—the focus has been on showing that the intricate dynamics leave the target posterior distribution invariant. Innovating in this arena requires constructing novel dynamics and simultaneously ensuring that the target distribution is the stationary distribution. This can be quite challenging, and often requires significant physical and geometrical intuition [6, 13, 7]. A natural question, then, is whether there exists a general recipe for devising such continuous-dynamic MCMC methods that naturally lead to invariance of the target distribution. In this paper, we answer this question to the affirmative. Furthermore, and quite importantly, our proposed recipe is *complete*. That is, any continuous Markov process (with no jumps) with the desired invariant distribution can be cast within our framework, including HMC, Riemann manifold HMC, SGLD, SGHMC, their recent variants, and any future developments in this area. That is, our method provides a unifying framework of past algorithms, as well as a practical tool for devising new samplers and testing the correctness of proposed samplers.

The recipe involves defining a (stochastic) system parameterized by two matrices: a positive semidefinite diffusion matrix, $\mathbf{D}(\mathbf{z})$, and a skew-symmetric curl matrix, $\mathbf{Q}(\mathbf{z})$, where $\mathbf{z} = (\theta, r)$ with $\theta$ our model parameters of interest and $r$ a set of auxiliary variables. The dynamics are then written explicitly in terms of the target stationary distribution and these two matrices. By varying the choices of $\mathbf{D}(\mathbf{z})$ and $\mathbf{Q}(\mathbf{z})$, we explore the space of MCMC methods that maintain the correct invariant distribution. We constructively prove the completeness of this framework by converting a general continuous Markov process into the proposed dynamic structure.

For any given $\mathbf{D}(\mathbf{z})$, $\mathbf{Q}(\mathbf{z})$, and target distribution, we provide practical algorithms for implementing either full-data or minibatch-based variants of the sampler. In Sec. 3.1, we cast many previous continuous-dynamic samplers in our framework, finding their $\mathbf{D}(\mathbf{z})$ and $\mathbf{Q}(\mathbf{z})$. We then show how these existing $\mathbf{D}(\mathbf{z})$ and $\mathbf{Q}(\mathbf{z})$ building blocks can be used to devise new samplers; we leave the question of exploring the space of $\mathbf{D}(\mathbf{z})$ and $\mathbf{Q}(\mathbf{z})$ well-suited to the structure of the target distribution as an interesting direction for future research. In Sec. 3.2 we demonstrate our ability to construct new and relevant samplers by proposing *stochastic gradient Riemann Hamiltonian Monte Carlo*, the existence of which was previously only speculated. We demonstrate the utility of this sampler on synthetic data and in a streaming Wikipedia analysis using latent Dirichlet allocation [5].

## 2   A Complete Stochastic Gradient MCMC Framework

We start with the standard MCMC goal of drawing samples from a target distribution, which we take to be the posterior $p(\theta|\mathcal{S})$ of model parameters $\theta \in \mathbb{R}^d$ given an observed dataset $\mathcal{S}$. Throughout, we assume i.i.d. data $\mathbf{x} \sim p(\mathbf{x}|\theta)$. We write $p(\theta|\mathcal{S}) \propto \exp(-U(\theta))$, with *potential function* $U(\theta) = -\sum_{\mathbf{x} \in \mathcal{S}} \log p(\mathbf{x}|\theta) - \log p(\theta)$. Algorithms like HMC [12, 10] further augment the space of interest with auxiliary variables $r$ and sample from $p(\mathbf{z}|\mathcal{S}) \propto \exp(-H(\mathbf{z}))$, with *Hamiltonian*

$$H(\mathbf{z}) = H(\theta, r) = U(\theta) + g(\theta, r), \quad \text{such that } \int \exp(-g(\theta, r))\mathrm{d}r = constant. \quad (1)$$

Marginalizing the auxiliary variables gives us the desired distribution on $\theta$. In this paper, we generically consider $\mathbf{z}$ as the samples we seek to draw; $\mathbf{z}$ could represent $\theta$ itself, or an augmented state space in which case we simply discard the auxiliary variables to perform the desired marginalization.

As in HMC, the idea is to translate the task of sampling from the posterior distribution to simulating from a continuous dynamical system which is used to define a Markov transition kernel. That is, over any interval $h$, the differential equation defines a mapping from the state at time $t$ to the state at time $t + h$. One can then discuss the evolution of the distribution $p(\mathbf{z}, t)$ under the dynamics, as characterized by the Fokker-Planck equation for stochastic dynamics [14] or the Liouville equation for deterministic dynamics [20]. This evolution can be used to analyze the *invariant distribution* of the dynamics, $p^s(\mathbf{z})$. When considering deterministic dynamics, as in HMC, a jump process must be added to ensure ergodicity. If the resulting stationary distribution is equal to the target posterior, then simulating from the process can be equated with drawing samples from the posterior.

If the stationary distribution is *not* the target distribution, a Metropolis-Hastings (MH) correction can often be applied. Unfortunately, such correction steps require a costly computation on the entire

dataset. Even if one can compute the MH correction, if the dynamics do not *nearly* lead to the correct stationary distribution, then the rejection rate can be high even for short simulation periods $h$. Furthermore, for many stochastic gradient MCMC samplers, computing the probability of the reverse path is infeasible, obviating the use of MH. As such, a focus in the literature is on defining dynamics with the right target distribution, especially in large-data scenarios where MH corrections are computationally burdensome or infeasible.

## 2.1 Devising SDEs with a Specified Target Stationary Distribution

Generically, all continuous Markov processes that one might consider for sampling can be written as a stochastic differential equation (SDE) of the form:

$$d\mathbf{z} = \mathbf{f}(\mathbf{z})dt + \sqrt{2\mathbf{D}(\mathbf{z})}d\mathbf{W}(t), \tag{2}$$

where $\mathbf{f}(\mathbf{z})$ denotes the deterministic drift and often relates to the gradient of $H(\mathbf{z})$, $\mathbf{W}(t)$ is a $d$-dimensional Wiener process, and $\mathbf{D}(\mathbf{z})$ is a positive semidefinite diffusion matrix. Clearly, however, not all choices of $\mathbf{f}(\mathbf{z})$ and $\mathbf{D}(\mathbf{z})$ yield the stationary distribution $p^s(\mathbf{z}) \propto \exp(-H(\mathbf{z}))$.

When $\mathbf{D}(\mathbf{z}) = 0$, as in HMC, the dynamics of Eq. (2) become deterministic. Our exposition focuses on SDEs, but our analysis applies to deterministic dynamics as well. In this case, our framework—using the Liouville equation in place of Fokker-Planck—ensures that the deterministic dynamics leave the target distribution invariant. For ergodicity, a jump process must be added, which is not considered in our recipe, but tends to be straightforward (e.g., momentum resampling in HMC).

To devise a recipe for constructing SDEs with the correct stationary distribution, we propose writing $\mathbf{f}(\mathbf{z})$ directly in terms of the target distribution:

$$\mathbf{f}(\mathbf{z}) = -\big[\mathbf{D}(\mathbf{z}) + \mathbf{Q}(\mathbf{z})\big]\nabla H(\mathbf{z}) + \Gamma(\mathbf{z}), \quad \Gamma_i(\mathbf{z}) = \sum_{j=1}^{d} \frac{\partial}{\partial \mathbf{z}_j}\big(\mathbf{D}_{ij}(\mathbf{z}) + \mathbf{Q}_{ij}(\mathbf{z})\big). \tag{3}$$

Here, $\mathbf{Q}(\mathbf{z})$ is a skew-symmetric curl matrix representing the deterministic traversing effects seen in HMC procedures. In contrast, the diffusion matrix $\mathbf{D}(\mathbf{z})$ determines the strength of the Wiener-process-driven diffusion. Matrices $\mathbf{D}(\mathbf{z})$ and $\mathbf{Q}(\mathbf{z})$ can be adjusted to attain faster convergence to the posterior distribution. A more detailed discussion on the interpretation of $\mathbf{D}(\mathbf{z})$ and $\mathbf{Q}(\mathbf{z})$ and the influence of specific choices of these matrices is provided in the Supplement.

Importantly, as we show in Theorem 1, sampling the stochastic dynamics of Eq. (2) (according to Itô integral) with $\mathbf{f}(\mathbf{z})$ as in Eq. (3) leads to the desired posterior distribution as the stationary distribution: $p^s(\mathbf{z}) \propto \exp(-H(\mathbf{z}))$. That is, for any choice of positive semidefinite $\mathbf{D}(\mathbf{z})$ and skew-symmetric $\mathbf{Q}(\mathbf{z})$ parameterizing $\mathbf{f}(\mathbf{z})$, we know that simulating from Eq. (2) will provide samples from $p(\theta \mid \mathcal{S})$ (discarding any sampled auxiliary variables $r$) assuming the process is ergodic.

**Theorem 1.** *$p^s(\mathbf{z}) \propto \exp(-H(\mathbf{z}))$ is a stationary distribution of the dynamics of Eq. (2) if $\mathbf{f}(\mathbf{z})$ is restricted to the form of Eq. (3), with $\mathbf{D}(\mathbf{z})$ positive semidefinite and $\mathbf{Q}(\mathbf{z})$ skew-symmetric. If $\mathbf{D}(\mathbf{z})$ is positive definite, or if ergodicity can be shown, then the stationary distribution is unique.*

*Proof.* The equivalence of $p^s(\mathbf{z})$ and the target $p(\mathbf{z} \mid \mathcal{S}) \propto \exp(-H(\mathbf{z}))$ can be shown using the Fokker-Planck description of the probability density evolution under the dynamics of Eq. (2) :

$$\partial_t p(\mathbf{z}, t) = -\sum_i \frac{\partial}{\partial \mathbf{z}_i}\big(\mathbf{f}_i(\mathbf{z})p(\mathbf{z}, t)\big) + \sum_{i,j} \frac{\partial^2}{\partial \mathbf{z}_i \partial \mathbf{z}_j}\big(\mathbf{D}_{ij}(\mathbf{z})p(\mathbf{z}, t)\big). \tag{4}$$

Eq. (4) can be further transformed into a more compact form [19, 16]:

$$\partial_t p(\mathbf{z}, t) = \nabla^T \cdot \Big([\mathbf{D}(\mathbf{z}) + \mathbf{Q}(\mathbf{z})]\left[p(\mathbf{z}, t)\nabla H(\mathbf{z}) + \nabla p(\mathbf{z}, t)\right]\Big). \tag{5}$$

We can verify that $p(\mathbf{z} \mid \mathcal{S})$ is invariant under Eq. (5) by calculating $\left[e^{-H(\mathbf{z})}\nabla H(\mathbf{z}) + \nabla e^{-H(\mathbf{z})}\right] = 0$. If the process is ergodic, this invariant distribution is unique. The equivalence of the compact form was originally proved in [16]; we include a detailed proof in the Supplement for completeness. $\square$

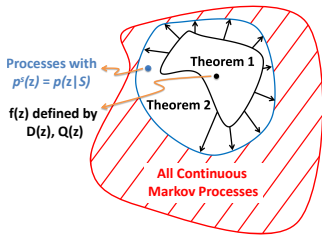

Figure 1: The red space represents the set of all continuous Markov processes. A point in the black space represents a continuous Markov process defined by Eqs. (2)-(3) based on a specific choice of $\mathbf{D}(\mathbf{z}), \mathbf{Q}(\mathbf{z})$. By Theorem 1, each such point has stationary distribution $p^s(\mathbf{z}) = p(\mathbf{z} \mid \mathcal{S})$. The blue space represents *all* continuous Markov processes with $p^s(\mathbf{z}) = p(\mathbf{z} \mid \mathcal{S})$. Theorem 2 states that these blue and black spaces are equivalent (there is no gap, and any point in the blue space has a corresponding $\mathbf{D}(\mathbf{z}), \mathbf{Q}(\mathbf{z})$ in our framework).

## 2.2 Completeness of the Framework

An important question is what portion of samplers defined by continuous Markov processes with the target invariant distribution can we define by iterating over all possible $\mathbf{D}(\mathbf{z})$ and $\mathbf{Q}(\mathbf{z})$? In Theorem 2, we show that for *any* continuous Markov process with the desired stationary distribution, $p^s(\mathbf{z})$, there exists an SDE as in Eq. (2) with $\mathbf{f}(\mathbf{z})$ defined as in Eq. (3). We know from the Chapman-Kolmogorov equation [9] that any continuous Markov process with stationary distribution $p^s(\mathbf{z})$ can be written as in Eq. (2), which gives us the diffusion matrix $\mathbf{D}(\mathbf{z})$. Theorem 2 then constructively defines the curl matrix $\mathbf{Q}(\mathbf{z})$. This result implies that our recipe is *complete*. That is, we cover *all* possible continuous Markov process samplers in our framework. See Fig. 1.

**Theorem 2.** *For the SDE of Eq.* (2)*, suppose its stationary probability density function $p^s(\mathbf{z})$ uniquely exists, and that* $\left[ \mathbf{f}_i(\mathbf{z})p^s(\mathbf{z}) - \sum_{j=1}^d \frac{\partial}{\partial \theta_j} \left( \mathbf{D}_{ij}(\mathbf{z})p^s(\mathbf{z}) \right) \right]$ *is integrable with respect to the Lebesgue measure, then there exists a skew-symmetric $\mathbf{Q}(\mathbf{z})$ such that Eq.* (3) *holds.*

The integrability condition is usually satisfied when the probability density function uniquely exists. A constructive proof for the existence of $\mathbf{Q}(\mathbf{z})$ is provided in the Supplement.

## 2.3 A Practical Algorithm

In practice, simulation relies on an $\epsilon$-discretization of the SDE, leading to a **full-data** update rule

$$\mathbf{z}_{t+1} \leftarrow \mathbf{z}_t - \epsilon_t \left[ \left( \mathbf{D}(\mathbf{z}_t) + \mathbf{Q}(\mathbf{z}_t) \right) \nabla H(\mathbf{z}_t) + \Gamma(\mathbf{z}_t) \right] + \mathcal{N}(0, 2\epsilon_t \mathbf{D}(\mathbf{z}_t)). \tag{6}$$

Calculating the gradient of $H(\mathbf{z})$ involves evaluating the gradient of $U(\theta)$. For a stochastic gradient method, the assumption is that $U(\theta)$ is too computationally intensive to compute as it relies on a sum over all data points (see Sec. 2). Instead, such stochastic gradient algorithms examine *independently sampled* data subsets $\widetilde{\mathcal{S}} \subset \mathcal{S}$ and the corresponding potential for these data:

$$\widetilde{U}(\theta) = -\frac{|\mathcal{S}|}{|\widetilde{\mathcal{S}}|} \sum_{\mathbf{x} \in \widetilde{\mathcal{S}}} \log p(\mathbf{x}|\theta) - \log p(\theta); \quad \widetilde{\mathcal{S}} \subset \mathcal{S}. \tag{7}$$

The specific form of Eq. (7) implies that $\widetilde{U}(\theta)$ is an unbiased estimator of $U(\theta)$. As such, a gradient computed based on $\widetilde{U}(\theta)$—called a *stochastic gradient* [15]—is a noisy, but unbiased estimator of the full-data gradient. The key question in many of the existing stochastic gradient MCMC algorithms is whether the noise injected by the stochastic gradient adversely affects the stationary distribution of the modified dynamics (using $\nabla \widetilde{U}(\theta)$ in place of $\nabla U(\theta)$). One way to analyze the impact of the stochastic gradient is to make use of the central limit theorem and assume

$$\nabla \widetilde{U}(\theta) = \nabla U(\theta) + \mathcal{N}(0, \mathbf{V}(\theta)), \tag{8}$$

resulting in a noisy Hamiltonian gradient $\nabla \widetilde{H}(\mathbf{z}) = \nabla H(\mathbf{z}) + [\mathcal{N}(0, \mathbf{V}(\theta)), \mathbf{0}]^T$. Simply plugging in $\nabla \widetilde{H}(\mathbf{z})$ in place of $\nabla H(\mathbf{z})$ in Eq. (6) results in dynamics with an additional noise term $(\mathbf{D}(\mathbf{z}_t) + \mathbf{Q}(\mathbf{z}_t))[\mathcal{N}(0, \mathbf{V}(\theta)), \mathbf{0}]^T$. To counteract this, assume we have an estimate $\hat{\mathbf{B}}_t$ of the variance of this additional noise satisfying $2\mathbf{D}(\mathbf{z}_t) - \epsilon_t \hat{\mathbf{B}}_t \succeq 0$ (i.e., positive semidefinite). With small $\epsilon$, this is always true since the stochastic gradient noise scales down faster than the added noise. Then, we can attempt to account for the stochastic gradient noise by simulating

$$\mathbf{z}_{t+1} \leftarrow \mathbf{z}_t - \epsilon_t \left[ \left( \mathbf{D}(\mathbf{z}_t) + \mathbf{Q}(\mathbf{z}_t) \right) \nabla \widetilde{H}(\mathbf{z}_t) + \Gamma(\mathbf{z}_t) \right] + \mathcal{N}(0, \epsilon_t(2\mathbf{D}(\mathbf{z}_t) - \epsilon_t \hat{\mathbf{B}}_t)). \tag{9}$$

This provides our **stochastic gradient**—or *minibatch*— variant of the sampler. In Eq. (9), the noise introduced by the stochastic gradient is multiplied by $\epsilon_t$ (and the compensation by $\epsilon_t^2$), implying that

the discrepancy between these dynamics and those of Eq. (6) approaches zero as $\epsilon_t$ goes to zero. As such, in this infinitesimal step size limit, since Eq. (6) yields the correct invariant distribution, so does Eq. (9). This avoids the need for a costly or potentially intractable MH correction. However, having to decrease $\epsilon_t$ to zero comes at the cost of increasingly small updates. We can also use a finite, small step size in practice, resulting in a biased (but faster) sampler. A similar bias-speed tradeoff was used in [11, 3] to construct MH samplers, in addition to being used in SGLD and SGHMC.

## 3 Applying the Theory to Construct Samplers

### 3.1 Casting Previous MCMC Algorithms within the Proposed Framework

We explicitly state how some recently developed MCMC methods fall within the proposed framework based on specific choices of $\mathbf{D}(\mathbf{z})$, $\mathbf{Q}(\mathbf{z})$ and $H(\mathbf{z})$ in Eq. (2) and (3). For the stochastic gradient methods, we show how our framework can be used to "reinvent" the samplers by guiding their construction and avoiding potential mistakes or inefficiencies caused by naïve implementations.

**Hamiltonian Monte Carlo (HMC)**   The key ingredient in HMC [8, 12] is Hamiltonian dynamics, which simulate the physical motion of an object with position $\theta$, momentum $r$, and mass $\mathbf{M}$ on an frictionless surface as follows (typically, a leapfrog simulation is used instead):

$$\begin{cases} \theta_{t+1} \leftarrow \theta_t + \epsilon_t \mathbf{M}^{-1} r_t \\ r_{t+1} \leftarrow r_t - \epsilon_t \nabla U(\theta_t). \end{cases} \tag{10}$$

Eq. (10) is a special case of the proposed framework with $\mathbf{z} = (\theta, r)$, $H(\theta, r) = U(\theta) + \frac{1}{2} r^T M^{-1} r$, $\mathbf{Q}(\theta, r) = \begin{pmatrix} 0 & -\mathbf{I} \\ \mathbf{I} & 0 \end{pmatrix}$ and $\mathbf{D}(\theta, r) = \mathbf{0}$.

**Stochastic Gradient Hamiltonian Monte Carlo (SGHMC)**   As discussed in [6], simply replacing $\nabla U(\theta)$ by the stochastic gradient $\nabla \widetilde{U}(\theta)$ in Eq. (10) results in the following updates:

$$\text{Naive}: \begin{cases} \theta_{t+1} \leftarrow \theta_t + \epsilon_t \mathbf{M}^{-1} r_t \\ r_{t+1} \leftarrow r_t - \epsilon_t \nabla \widetilde{U}(\theta_t) \approx r_t - \epsilon_t \nabla U(\theta_t) + \mathcal{N}(0, \epsilon_t^2 \mathbf{V}(\theta_t)), \end{cases} \tag{11}$$

where the $\approx$ arises from the approximation of Eq. (8). Careful study shows that Eq. (11) *cannot* be rewritten into our proposed framework, which hints that such a naïve stochastic gradient version of HMC is not correct. Interestingly, the authors of [6] proved that this naïve version indeed does not have the correct stationary distribution. In our framework, we see that the noise term $\mathcal{N}(0, 2\epsilon_t \mathbf{D}(\mathbf{z}))$ is paired with a $\mathbf{D}(\mathbf{z})\nabla H(\mathbf{z})$ term, hinting that such a term should be added to Eq. (11). Here, $\mathbf{D}(\theta, r) = \begin{pmatrix} 0 & 0 \\ 0 & \epsilon \mathbf{V}(\theta) \end{pmatrix}$, which means we need to add $\mathbf{D}(\mathbf{z})\nabla H(\mathbf{z}) = \epsilon \mathbf{V}(\theta)\nabla_r H(\theta, r) = \epsilon \mathbf{V}(\theta)\mathbf{M}^{-1} r$. Interestingly, this is the correction strategy proposed in [6], but through a physical interpretation of the dynamics. In particular, the term $\epsilon \mathbf{V}(\theta)\mathbf{M}^{-1} r$ (or, generically, $\mathbf{C}\mathbf{M}^{-1} r$ where $\mathbf{C} \succeq \epsilon \mathbf{V}(\theta)$) has an interpretation as friction and leads to second order Langevin dynamics:

$$\begin{cases} \theta_{t+1} \leftarrow \theta_t + \epsilon_t \mathbf{M}^{-1} r_t \\ r_{t+1} \leftarrow r_t - \epsilon_t \nabla \widetilde{U}(\theta_t) - \epsilon_t \mathbf{C} \mathbf{M}^{-1} r_t + \mathcal{N}(0, \epsilon_t(2\mathbf{C} - \epsilon_t \hat{\mathbf{B}}_t)). \end{cases} \tag{12}$$

Here, $\hat{\mathbf{B}}_t$ is an estimate of $\mathbf{V}(\theta_t)$. This method now fits into our framework with $H(\theta, r)$ and $\mathbf{Q}(\theta, r)$ as in HMC, but with $\mathbf{D}(\theta, r) = \begin{pmatrix} 0 & 0 \\ 0 & \mathbf{C} \end{pmatrix}$. This example shows how our theory can be used to identify invalid samplers and provide guidance on how to effortlessly correct the mistakes; this is crucial when physical intuition is not available. Once the proposed sampler is cast in our framework with a specific $\mathbf{D}(\mathbf{z})$ and $\mathbf{Q}(\mathbf{z})$, there is no need for sampler-specific proofs, such as those of [6].

**Stochastic Gradient Langevin Dynamics (SGLD)**   SGLD [17] proposes to use the following first order (no momentum) Langevin dynamics to generate samples

$$\theta_{t+1} \leftarrow \theta_t - \epsilon_t \mathbf{D} \nabla \widetilde{U}(\theta_t) + \mathcal{N}(0, 2\epsilon_t \mathbf{D}). \tag{13}$$

This algorithm corresponds to taking $\mathbf{z} = \theta$ with $H(\theta) = U(\theta), \mathbf{D}(\theta) = \mathbf{D}, \mathbf{Q}(\theta) = \mathbf{0}$, and $\hat{\mathbf{B}}_t = \mathbf{0}$. As motivated by Eq. (9) of our framework, the variance of the stochastic gradient can be subtracted from the sampler injected noise to make the finite stepsize simulation more accurate. This variant of SGLD leads to the stochastic gradient Fisher scoring algorithm [1].

**Stochastic Gradient Riemannian Langevin Dynamics (SGRLD)** SGLD can be generalized to use an adaptive diffusion matrix $\mathbf{D}(\theta)$. Specifically, it is interesting to take $\mathbf{D}(\theta) = \mathbf{G}^{-1}(\theta)$, where $\mathbf{G}(\theta)$ is the Fisher information metric. The sampler dynamics are given by

$$\theta_{t+1} \leftarrow \theta_t - \epsilon_t[\mathbf{G}(\theta_t)^{-1}\nabla\widetilde{U}(\theta_t) + \Gamma(\theta_t)] + \mathcal{N}(0, 2\epsilon_t\mathbf{G}(\theta_t)^{-1}). \qquad (14)$$

Taking $\mathbf{D}(\theta) = \mathbf{G}(\theta)^{-1}$, $\mathbf{Q}(\theta) = \mathbf{0}$, and $\hat{\mathbf{B}}_t = \mathbf{0}$, this SGRLD [13] method falls into our framework with correction term $\Gamma_i(\theta) = \sum_j \dfrac{\partial \mathbf{D}_{ij}(\theta)}{\partial \theta_j}$. It is interesting to note that in earlier literature [10], $\Gamma_i(\theta)$ was taken to be $2\,|\mathbf{G}(\theta)|^{-1/2} \sum_j \dfrac{\partial}{\partial \theta_j}\left(\mathbf{G}_{ij}^{-1}(\theta)|\mathbf{G}(\theta)|^{1/2}\right)$. More recently, it was found that this correction term corresponds to the distribution function with respect to a non-Lebesgue measure [18]; for the Lebesgue measure, the revised $\Gamma_i(\theta)$ was as determined by our framework [18]. Again, we have an example of our theory providing guidance in devising correct samplers.

**Stochastic Gradient Nosé-Hoover Thermostat (SGNHT)** Finally, the SGNHT [7] method incorporates ideas from thermodynamics to further increase adaptivity by augmenting the SGHMC system with an additional scalar auxiliary variable, $\xi$. The algorithm uses the following dynamics:

$$\begin{cases} \theta_{t+1} \leftarrow \theta_t + \epsilon_t r_t \\ r_{t+1} \leftarrow r_t - \epsilon_t \nabla\widetilde{U}(\theta_t) - \epsilon_t \xi_t r_t + \mathcal{N}(0, \epsilon_t(2A - \epsilon_t\hat{\mathbf{B}}_t)) \\ \xi_{t+1} \leftarrow \xi_t + \epsilon_t \left(\dfrac{1}{d}r_t^T r_t - 1\right). \end{cases} \qquad (15)$$

We can take $\mathbf{z} = (\theta, r, \xi)$, $H(\theta, r, \xi) = U(\theta) + \dfrac{1}{2}r^T r + \dfrac{1}{2d}(\xi - A)^2$, $\mathbf{D}(\theta, r, \xi) = \begin{pmatrix} 0 & 0 & 0 \\ 0 & A \cdot \mathbf{I} & 0 \\ 0 & 0 & 0 \end{pmatrix}$,

and $\mathbf{Q}(\theta, r, \xi) = \begin{pmatrix} 0 & -\mathbf{I} & 0 \\ \mathbf{I} & 0 & r/d \\ 0 & -r^T/d & 0 \end{pmatrix}$ to place these dynamics within our framework.

**Summary** In our framework, SGLD and SGRLD take $\mathbf{Q}(\mathbf{z}) = 0$ and instead stress the design of the diffusion matrix $\mathbf{D}(\mathbf{z})$, with SGLD using a constant $\mathbf{D}(\mathbf{z})$ and SGRLD an adaptive, $\theta$-dependent diffusion matrix to better account for the geometry of the space being explored. On the other hand, HMC takes $\mathbf{D}(\mathbf{z}) = 0$ and focuses on the curl matrix $\mathbf{Q}(\mathbf{z})$. SGHMC combines SGLD with HMC through non-zero $\mathbf{D}(\theta)$ and $\mathbf{Q}(\theta)$ matrices. SGNHT then extends SGHMC by taking $\mathbf{Q}(\mathbf{z})$ to be state dependent. The relationships between these methods are depicted in the Supplement, which likewise contains a discussion of the tradeoffs between these two matrices. In short, $\mathbf{D}(\mathbf{z})$ can guide escaping from local modes while $\mathbf{Q}(\mathbf{z})$ can enable rapid traversing of low-probability regions, especially when state adaptation is incorporated. We readily see that most of the product space $\mathbf{D}(\mathbf{z}) \times \mathbf{Q}(\mathbf{z})$, defining the space of all possible samplers, has yet to be filled.

### 3.2 Stochastic Gradient Riemann Hamiltonian Monte Carlo

In Sec. 3.1, we have shown how our framework unifies existing samplers. In this section, we now use our framework to guide the development of a new sampler. While SGHMC [6] inherits the momentum term of HMC, making it easier to traverse the space of parameters, the underlying geometry of the target distribution is still not utilized. Such information can usually be represented by the Fisher information metric [10], denoted as $\mathbf{G}(\theta)$, which can be used to precondition the dynamics. For our proposed system, we consider $H(\theta, r) = U(\theta) + \frac{1}{2}r^T r$, as in HMC/SGHMC methods, and modify the $\mathbf{D}(\theta, r)$ and $\mathbf{Q}(\theta, r)$ of SGHMC to account for the geometry as follows:

$$\mathbf{D}(\theta, r) = \begin{pmatrix} 0 & 0 \\ 0 & \mathbf{G}(\theta)^{-1} \end{pmatrix}; \qquad \mathbf{Q}(\theta, r) = \begin{pmatrix} 0 & -\mathbf{G}(\theta)^{-1/2} \\ \mathbf{G}(\theta)^{-1/2} & 0 \end{pmatrix}.$$

We refer to this algorithm as *stochastic gradient Riemann Hamiltonian Monte Carlo* (**SGRHMC**). Our theory holds for any positive definite $\mathbf{G}(\theta)$, yielding a *generalized SGRHMC* (**gSGRHMC**) algorithm, which can be helpful when the Fisher information metric is hard to compute.

A naïve implementation of a state-dependent SGHMC algorithm might simply (i) precondition the HMC update, (ii) replace $\nabla U(\theta)$ by $\nabla\widetilde{U}(\theta)$, and (iii) add a state-dependent friction term on the order of the diffusion matrix to counterbalance the noise as in SGHMC, resulting in:

$$\text{Naïve}: \begin{cases} \theta_{t+1} \leftarrow & \theta_t + \epsilon_t \mathbf{G}(\theta_t)^{-1/2} r_t \\ r_{t+1} \leftarrow & r_t - \epsilon_t \mathbf{G}(\theta_t)^{-1/2}\nabla_\theta\widetilde{U}(\theta_t) - \epsilon_t \mathbf{G}(\theta_t)^{-1} r_t + \mathcal{N}(0, \epsilon_t(2\mathbf{G}(\theta_t)^{-1} - \epsilon_t\hat{\mathbf{B}}_t)). \end{cases} \qquad (16)$$

**Algorithm 1:** Generalized Stochastic Gradient Riemann Hamiltonian Monte Carlo

initialize $(\theta_0, r_0)$
**for** $t = 0, 1, 2 \cdots$ **do**
$\quad$ optionally, periodically resample momentum $r$ as $r^{(t)} \sim \mathcal{N}(0, \mathbf{I})$
$\quad \theta_{t+1} \leftarrow \theta_t + \epsilon_t \mathbf{G}(\theta_t)^{-1/2} r_t, \quad \Sigma_t \leftarrow \epsilon_t (2\mathbf{G}(\theta_t)^{-1} - \epsilon_t \hat{\mathbf{B}}_t)$
$\quad r_{t+1} \leftarrow r_t - \epsilon_t \mathbf{G}(\theta_t)^{-1/2} \nabla_\theta \widetilde{U}(\theta_t) + \epsilon_t \nabla_\theta (\mathbf{G}(\theta_t)^{-1/2}) - \epsilon_t \mathbf{G}(\theta_t)^{-1} r_t + \mathcal{N}\left(0, \Sigma_t\right)$
**end**

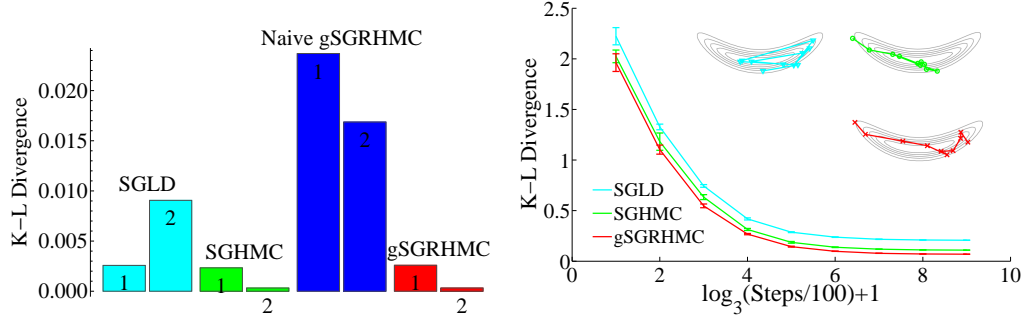

Figure 2: *Left:* For two simulated 1D distributions defined by $U(\theta) = \theta^2/2$ (*one peak*) and $U(\theta) = \theta^4 - 2\theta^2$ (*two peaks*), we compare the KL divergence of methods: SGLD, SGHMC, the naïve SGRHMC of Eq. (16), and the gSGRHMC of Eq. (17) relative to the true distribution in each scenario (left and right bars labeled by 1 and 2). *Right:* For a correlated 2D distribution with $U(\theta_1, \theta_2) = \theta_1^4/10 + (4 \cdot (\theta_2 + 1.2) - \theta_1^2)^2/2$, we see that our gSGRHMC most rapidly explores the space relative to SGHMC and SGLD. Contour plots of the distribution along with paths of the first 10 sampled points are shown for each method.

However, as we show in Sec. 4.1, samples from these dynamics do not converge to the desired distribution. Indeed, this system cannot be written within our framework. Instead, we can simply follow our framework and, as indicated by Eq. (9), consider the following update rule:

$$\begin{cases} \theta_{t+1} \leftarrow \theta_t + \epsilon_t \mathbf{G}(\theta_t)^{-1/2} r_t \\ r_{t+1} \leftarrow r_t - \epsilon_t [\mathbf{G}(\theta)^{-1/2} \nabla_\theta \widetilde{U}(\theta_t) + \nabla_\theta \left(\mathbf{G}(\theta_t)^{-1/2}\right) - \mathbf{G}(\theta_t)^{-1} r_t] + \mathcal{N}(0, \epsilon_t (2\mathbf{G}(\theta_t)^{-1} - \epsilon_t \hat{\mathbf{B}}_t)), \end{cases}$$
(17)

which includes a correction term $\nabla_\theta \left(\mathbf{G}(\theta)^{-1/2}\right)$, with $i$-th component $\sum_j \frac{\partial}{\partial \theta_j} \left(\mathbf{G}(\theta)^{-1/2}\right)_{ij}$. The practical implementation of gSGRHMC is outlined in Algorithm 1.

## 4 Experiments

In Sec. 4.1, we show that gSGRHMC can excel at rapidly exploring distributions with complex landscapes. We then apply SGRHMC to sampling in a latent Dirichlet allocation (LDA) model on a large Wikipedia dataset in Sec. 4.2. The Supplement contains details on the specific samplers considered and the parameter settings used in these experiments.

### 4.1 Synthetic Experiments

In this section we aim to empirically (i) validate the correctness of our recipe and (ii) assess the effectiveness of gSGRHMC. In Fig. 2(left), we consider two univariate distributions (shown in the Supplement) and compare SGLD, SGHMC, the naïve state-adaptive SGHMC of Eq. (16), and our proposed gSGRHMC of Eq. (17). See the Supplement for the form of $\mathbf{G}(\theta)$. As expected, the naïve implementation does not converge to the target distribution. In contrast, the gSGRHMC algorithm obtained via our recipe indeed has the correct invariant distribution and efficiently explores the distributions. In the second experiment, we sample a bivariate distribution with strong correlation. The results are shown in Fig. 2(right). The comparison between SGLD, SGHMC, and our gSGRHMC method shows that both a state-dependent preconditioner and Hamiltonian dynamics help to make the sampler more efficient than either element on its own.

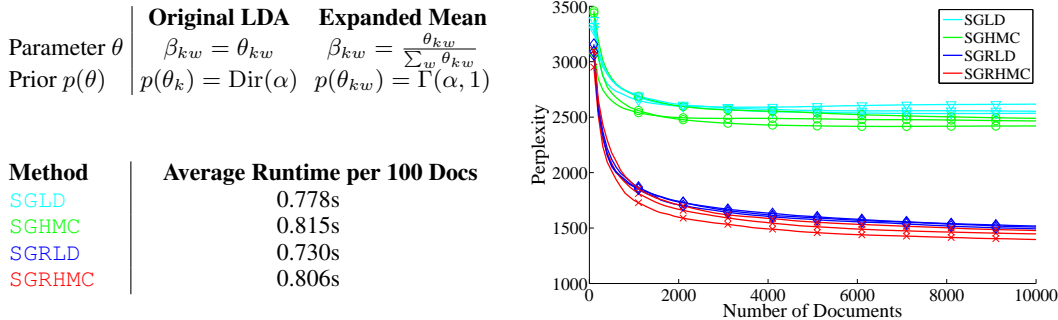

| Parameter $\theta$ | Original LDA | Expanded Mean |
|---|---|---|
| | $\beta_{kw} = \theta_{kw}$ | $\beta_{kw} = \frac{\theta_{kw}}{\sum_w \theta_{kw}}$ |
| Prior $p(\theta)$ | $p(\theta_k) = \mathrm{Dir}(\alpha)$ | $p(\theta_{kw}) = \Gamma(\alpha, 1)$ |

| Method | Average Runtime per 100 Docs |
|---|---|
| SGLD | 0.778s |
| SGHMC | 0.815s |
| SGRLD | 0.730s |
| SGRHMC | 0.806s |

Figure 3: *Upper Left:* Expanded mean parameterization of the LDA model. *Lower Left:* Average runtime per 100 Wikipedia entries for all methods. *Right:* Perplexity versus number of Wikipedia entries processed.

## 4.2 Online Latent Dirichlet Allocation

We also applied SGRHMC (with $\mathbf{G}(\theta) = \mathrm{diag}(\theta)^{-1}$, the Fisher information metric) to an *online* latent Dirichlet allocation (LDA) [5] analysis of topics present in Wikipedia entries. In LDA, each topic is associated with a distribution over words, with $\beta_{kw}$ the probability of word $w$ under topic $k$. Each document is comprised of a mixture of topics, with $\pi_k^{(d)}$ the probability of topic $k$ in document $d$. Documents are generated by first selecting a topic $z_j^{(d)} \sim \pi^{(d)}$ for the $j$th word and then drawing the specific word from the topic as $x_j^{(d)} \sim \beta_{z_j^{(d)}}$. Typically, $\pi^{(d)}$ and $\beta_k$ are given Dirichlet priors.

The goal of our analysis here is inference of the corpus-wide topic distributions $\beta_k$. Since the Wikipedia dataset is large and continually growing with new articles, it is not practical to carry out this task over the whole dataset. Instead, we scrape the corpus from Wikipedia in a streaming manner and sample parameters based on minibatches of data. Following the approach in [13], we first analytically marginalize the document distributions $\pi^{(d)}$ and, to resolve the boundary issue posed by the Dirichlet posterior of $\beta_k$ defined on the probability simplex, use an expanded mean parameterization shown in Figure 3(upper left). Under this parameterization, we then compute $\nabla \log p(\theta|\mathbf{x})$ and, in our implementation, use boundary reflection to ensure the positivity of parameters $\theta_{kw}$. The necessary expectation over word-specific topic indicators $z_j^{(d)}$ is approximated using Gibbs sampling separately on each document, as in [13]. The Supplement contains further details.

For all the methods, we report results of three random runs. When sampling distributions with mass concentrated over small regions, as in this application, it is important to incorporate geometric information via a Riemannian sampler [13]. The results in Fig. 3(right) indeed demonstrate the importance of Riemannian variants of the stochastic gradient samplers. However, there also appears to be some benefits gained from the incorporation of the HMC term for both the Riemmannian and non-Reimannian samplers. The average runtime for the different methods are similar (see Fig. 3(lower left)) since the main computational bottleneck is the gradient evaluation. Overall, this application serves as an important example of where our newly proposed sampler can have impact.

## 5 Conclusion

We presented a general recipe for devising MCMC samplers based on continuous Markov processes. Our framework constructs an SDE specified by two matrices, a positive semidefinite $\mathbf{D}(\mathbf{z})$ and a skew-symmetric $\mathbf{Q}(\mathbf{z})$. We prove that for any $\mathbf{D}(\mathbf{z})$ and $\mathbf{Q}(\mathbf{z})$, we can devise a continuous Markov process with a specified stationary distribution. We also prove that for any continuous Markov process with the target stationary distribution, there exists a $\mathbf{D}(\mathbf{z})$ and $\mathbf{Q}(\mathbf{z})$ that cast the process in our framework. Our recipe is particularly useful in the more challenging case of devising stochastic gradient MCMC samplers. We demonstrate the utility of our recipe in "reinventing" previous stochastic gradient MCMC samplers, and in proposing our SGRHMC method. The efficiency and scalability of the SGRHMC method was shown on simulated data and a streaming Wikipedia analysis.

### Acknowledgments

This work was supported in part by ONR Grant N00014-10-1-0746, NSF CAREER Award IIS-1350133, and the TerraSwarm Research Center sponsored by MARCO and DARPA. We also thank Mr. Lei Wu for helping with the proof of Theorem 2 and Professors Ping Ao and Hong Qian for many discussions.

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
