[Supplementary Material]

# Supplementary Material: A Complete Recipe for Stochastic Gradient MCMC

**Yi-An Ma**, **Tianqi Chen**, and **Emily B. Fox**

University of Washington {yianma@u,tqchen@cs,ebfox@stat}.washington.edu

## 1 Proof of Stationary Distribution

In this section, we provide a proof for Theorem 1. To prove the theorem, we first show when $\mathbf{f}(\mathbf{z})$ satisfies Eq. (3) in the main paper, then the following Fokker-Planck equation of the dynamics:

$$\partial_t p(\mathbf{z}, t) = -\sum_i \frac{\partial}{\partial \mathbf{z}_i} \big(\mathbf{f}_i(\mathbf{z}) p(\mathbf{z}, t)\big) + \sum_{i,j} \frac{\partial^2}{\partial \mathbf{z}_i \partial \mathbf{z}_j} \big(\mathbf{D}_{ij}(\mathbf{z}) p(\mathbf{z}, t)\big),$$

is equivalent to the following compact form [7, 5]:

$$\partial_t p(\mathbf{z}, t) = \nabla^T \cdot \Big( [\mathbf{D}(\mathbf{z}) + \mathbf{Q}(\mathbf{z})] \left[ p(\mathbf{z}, t) \nabla H(\mathbf{z}) + \nabla p(\mathbf{z}, t) \right] \Big). \tag{S.1}$$

*Proof.* The proof is a re-writing of Eq. (S.1):

$$\partial_t p(\mathbf{z}, t) = \nabla^T \cdot \Big( [\mathbf{D}(\mathbf{z}) + \mathbf{Q}(\mathbf{z})] \left[ p(\mathbf{z}, t) \nabla H(\mathbf{z}) + \nabla p(\mathbf{z}, t) \right] \Big)$$

$$= \sum_{i=1} \frac{\partial}{\partial \mathbf{z}_i} \sum_j [\mathbf{D}_{ij}(\mathbf{z}) + \mathbf{Q}_{ij}(\mathbf{z})] \left[ p(\mathbf{z}, t) \frac{\partial}{\partial \mathbf{z}_j} H(\mathbf{z}) + \frac{\partial}{\partial \mathbf{z}_j} p(\mathbf{z}, t) \right]$$

$$= \sum_i \frac{\partial}{\partial \mathbf{z}_i} \left\{ \sum_j [\mathbf{D}_{ij}(\mathbf{z}) + \mathbf{Q}_{ij}(\mathbf{z})] \, p(\mathbf{z}, t) \frac{\partial}{\partial \mathbf{z}_j} H(\mathbf{z}) \right\} + \sum_i \frac{\partial}{\partial \mathbf{z}_i} [\mathbf{D}_{ij}(\mathbf{z}) + \mathbf{Q}_{ij}(\mathbf{z})] \sum_j \frac{\partial}{\partial \mathbf{z}_j} p(\mathbf{z}, t).$$

We can further decompose the second term as follows

$$\sum_i \frac{\partial}{\partial \mathbf{z}_i} \mathbf{D}_{ij}(\mathbf{z}) \sum_j \frac{\partial}{\partial \mathbf{z}_j} p(\mathbf{z}, t) = \sum_{ij} \frac{\partial}{\partial \mathbf{z}_i} \frac{\partial}{\partial \mathbf{z}_j} [\mathbf{D}_{ij}(\mathbf{z}) p(\mathbf{z}, t)] - \sum_i \frac{\partial}{\partial \mathbf{z}_i} \left\{ p(\mathbf{z}, t) \left[ \sum_j \frac{\partial}{\partial \mathbf{z}_j} \mathbf{D}_{ij}(\mathbf{z}) \right] \right\}$$

$$\sum_i \frac{\partial}{\partial \mathbf{z}_i} \mathbf{Q}_{ij}(\mathbf{z}) \sum_j \frac{\partial}{\partial \mathbf{z}_j} p(\mathbf{z}, t) = -\sum_i \frac{\partial}{\partial \mathbf{z}_i} \left\{ p(\mathbf{z}, t) \left[ \sum_j \frac{\partial}{\partial \mathbf{z}_j} \mathbf{Q}_{ij}(\mathbf{z}) \right] \right\}.$$

The second equality follows because $\sum_{ij} \frac{\partial}{\partial \mathbf{z}_i} \frac{\partial}{\partial \mathbf{z}_j} [\mathbf{Q}_{ij}(\mathbf{z}) p(\mathbf{z}, t)] = 0$ due to anti-symmetry of $\mathbf{Q}$.

Putting these back into the formula, we get

$$\partial_t p(\mathbf{z}, t) = \sum_i \frac{\partial}{\partial \mathbf{z}_i} \left\{ \left[ \sum_j [\mathbf{D}_{ij}(\mathbf{z}) + \mathbf{Q}_{ij}(\mathbf{z})] \frac{\partial}{\partial \mathbf{z}_j} H(\mathbf{z}) - \sum_j [\frac{\partial}{\partial \mathbf{z}_j} \mathbf{D}_{ij}(\mathbf{z}) + \frac{\partial}{\partial \mathbf{z}_j} \mathbf{Q}_{ij}(\mathbf{z})] \right] p(\mathbf{z}, t) \right\}$$

$$+ \sum_{ij} \frac{\partial^2}{\partial \mathbf{z}_i \partial \mathbf{z}_j} [\mathbf{D}_{ij}(\mathbf{z}) p(\mathbf{z}, t)]$$

$$= -\sum_i \frac{\partial}{\partial \mathbf{z}_i} \big(\mathbf{f}_i(\mathbf{z}) p(\mathbf{z}, t)\big) + \sum_{i,j} \frac{\partial^2}{\partial \mathbf{z}_i \partial \mathbf{z}_j} \big(\mathbf{D}_{ij}(\mathbf{z}) p(\mathbf{z}, t)\big).$$

$\square$

We can then verify that $p(\mathbf{z} \mid \mathcal{S}) \propto e^{-H(\mathbf{z})}$ is invariant under the compact form by calculating

$$[p(\mathbf{z}, t)\nabla H(\mathbf{z}) + \nabla p(\mathbf{z}, t)] \propto \left[ e^{-H(\mathbf{z})}\nabla H(\mathbf{z}) + \nabla e^{-H(\mathbf{z})} \right] = 0.$$

This completes the proof of Theorem 1.

The above proof follows directly from [5] and is provided here for readers' convenience.

## 2 Proof of Completeness

In this section, we provide a constructive proof for Theorem 2, the existence of $\mathbf{Q}(\mathbf{z})$.

The proof is first outlined as follows:

- We first rewrite Eq. (3) in the main paper and notice that finding matrix $\mathbf{Q}(\mathbf{z})$ is equivalent to finding the matrix $\mathbf{Q}(\mathbf{z})p^s(\mathbf{z})$ such that

$$\sum_j \frac{\partial}{\partial \mathbf{z}_j}\left( \mathbf{Q}_{ij}(\mathbf{z})p^s(\mathbf{z}) \right) = \mathbf{f}_i(\mathbf{z})p^s(\mathbf{z}) - \sum_j \frac{\partial}{\partial \mathbf{z}_j}\left( \mathbf{D}_{ij}(\mathbf{z})p^s(\mathbf{z}) \right),$$

where the right hand side is a divergence-free vector.

- We transform the above equation and its constraint into the frequency domain and obtain a set of linear equations.

- Then we construct a solution to the linear equations and use inverse Fourier transform to obtain $\mathbf{Q}(\mathbf{z})$.

The complete procedure is:

*Proof.* Multiplying $p^s(\mathbf{z})$ on both sides of Eq. (3) in the main paper, and noting that:

$$p^s(\mathbf{z}) = \exp\left( -H(\mathbf{z}) \right), \tag{S.2}$$

we arrive at:

$$\mathbf{f}_i(\mathbf{z})p^s(\mathbf{z}) = \sum_j \frac{\partial}{\partial \mathbf{z}_j}\left( (\mathbf{D}_{ij}(\mathbf{z}) + \mathbf{Q}_{ij}(\mathbf{z}))p^s(\mathbf{z}) \right). \tag{S.3}$$

The equation for $\mathbf{Q}_{ij}(\mathbf{z})$ can now be written as:

$$\sum_j \frac{\partial}{\partial \mathbf{z}_j}\left( \mathbf{Q}_{ij}(\mathbf{z})p^s(\mathbf{z}) \right) = \mathbf{f}_i(\mathbf{z})p^s(\mathbf{z}) - \sum_j \frac{\partial}{\partial \mathbf{z}_j}\left( \mathbf{D}_{ij}(\mathbf{z})p^s(\mathbf{z}) \right). \tag{S.4}$$

Recall that the Fokker-Planck equation for the stochastic process, Eq. (2), is:

$$\frac{\partial p(\mathbf{z}, t)}{\partial t} = -\nabla^T \cdot \left( \mathbf{f}(\mathbf{z})p(\mathbf{z}, t) \right) + \nabla^2 : \left( \mathbf{D}(\mathbf{z})p(\mathbf{z}, t) \right)$$

$$= -\sum_i \frac{\partial}{\partial \mathbf{z}_i}\left\{ \mathbf{f}_i(\mathbf{z})p(\mathbf{z}, t) - \sum_j \frac{\partial}{\partial \mathbf{z}_j}\left( \mathbf{D}_{ij}(\mathbf{z})p(\mathbf{z}, t) \right) \right\}. \tag{S.5}$$

We can immediately observe that the right hand side of Eq. (S.4) has a divergenceless property by substituting the stationary probability density function $p^s(\mathbf{z})$ into Eq. (S.5):

$$\sum_i \frac{\partial}{\partial \mathbf{z}_i}\left\{ \mathbf{f}_i(\mathbf{z})p^s(\mathbf{z}) - \sum_j \frac{\partial}{\partial \mathbf{z}_j}\left( \mathbf{D}_{ij}(\mathbf{z})p^s(\mathbf{z}) \right) \right\} = 0. \tag{S.6}$$

The nice forms of Eqs. (S.4) and (S.6) imply that the questions can be transformed into a linear algebra problem once we apply a Fourier transform to them. Denote the Fourier transform of

$\mathbf{Q}(\mathbf{z})p^s(\mathbf{z})$ as $\hat{\mathbf{Q}}(\mathbf{k})$; and Fourier transform of $\mathbf{f}_i(\mathbf{z})p^s(\mathbf{z}) - \sum_j \dfrac{\partial}{\partial \mathbf{z}_j}\left(\mathbf{D}_{ij}(\mathbf{z})p^s(\mathbf{z})\right)$ as $\hat{\mathbf{F}}_i(\mathbf{k})$, where $\mathbf{k} = (\mathbf{k}_1, \cdots, \mathbf{k}_n)^T$ is the set of the spectral variables. That is:

$$\hat{\mathbf{Q}}_{ij}(\mathbf{k}) = \int_{\mathcal{D}} \mathbf{Q}_{ij}(\mathbf{z})p^s(\mathbf{z})e^{-2\pi\mathrm{i}\,\mathbf{k}^T\mathbf{z}}\mathrm{d}\mathbf{z};$$

$$\hat{\mathbf{F}}_i(\mathbf{k}) = \int_{\mathcal{D}} \left(\mathbf{f}_i(\mathbf{z})p^s(\mathbf{z}) - \sum_j \frac{\partial}{\partial \mathbf{z}_j}\left(\mathbf{D}_{ij}(\mathbf{z})p^s(\mathbf{z})\right)\right)e^{-2\pi\mathrm{i}\,\mathbf{k}^T\mathbf{z}}\mathrm{d}\mathbf{z}.$$

Then, $\dfrac{\partial}{\partial \mathbf{z}_j}\left(\mathbf{Q}_{ij}(\mathbf{z})p^s(\mathbf{z})\right)$ is transformed to $2\pi\mathrm{i}\,\hat{\mathbf{Q}}_{ij}\mathbf{k}_j$, and Eq. (S.4) becomes the following equivalent form in Fourier space:

$$\begin{cases} 2\pi\mathrm{i}\,\hat{\mathbf{Q}}\mathbf{k} = \hat{\mathbf{F}} \\ \mathbf{k}^T\hat{\mathbf{F}} = 0. \end{cases} \tag{S.7}$$

Hence, it is clear that matrix $\hat{\mathbf{Q}}$ must be a skew-symmetric projection matrix from the span of $\mathbf{k}$ to the span of $\hat{\mathbf{F}}$, where $\mathbf{k}$ and $\hat{\mathbf{F}}$ are always orthogonal to each other. We thereby construct $\hat{\mathbf{Q}}$ as combination of two rank 1 projection matrices:

$$\hat{\mathbf{Q}} = (2\pi\mathrm{i})^{-1}\frac{\hat{\mathbf{F}}\mathbf{k}^T}{\mathbf{k}^T\mathbf{k}} - (2\pi\mathrm{i})^{-1}\frac{\mathbf{k}\hat{\mathbf{F}}^T}{\mathbf{k}^T\mathbf{k}}. \tag{S.8}$$

We arrive at the final result that matrix $\mathbf{Q}(\mathbf{z})$ is equal to $p^s(\mathbf{z})^{-1}$ times the inverse Fourier transform of $\hat{\mathbf{Q}}(\mathbf{k})$:

$$\mathbf{Q}_{ij}(\mathbf{z}) = p^s(\mathbf{z})^{-1}\int_{\mathcal{D}} \frac{\mathbf{k}_j\hat{\mathbf{F}}_i(\mathbf{k}) - \mathbf{k}_i\hat{\mathbf{F}}_j(\mathbf{k})}{(2\pi\mathrm{i})\cdot\sum_l \mathbf{k}_l^2}e^{2\pi\mathrm{i}\sum_l \mathbf{k}_l\mathbf{x}_l}\mathrm{d}\mathbf{k}. \tag{S.9}$$

Thus, if $\left(\mathbf{f}_i(\mathbf{z})p^s(\mathbf{z}) - \sum_j \dfrac{\partial}{\partial \mathbf{z}_j}\left(\mathbf{D}_{ij}(\mathbf{z})p^s(\mathbf{z})\right)\right)$ belongs to the space of $L^1$, then any continuous time Markov process, Eq. (2), can be turned into this new formulation. $\qquad \square$

**Remark 1.** *Entries in the skew-symmetric projector* $\mathbf{Q}_{ij}(\mathbf{z})$ *constructed here are real.*

*Denote* $\mathbf{a}_i^2 = \sum\limits_{l\neq i} \mathbf{k}_l^2$*, then the inverse Fourier transform of* $\dfrac{\mathbf{k}_i}{(2\pi\mathrm{i})\cdot\sum\limits_l \mathbf{k}_l^2}$ *along the partial variable* $\mathbf{k}_i$ *is equal to:*

$$\mathbf{g}_i(\mathbf{z}) = -\frac{1}{2}e^{-2\pi\mathbf{a}_i\mathbf{z}_i}H[\mathbf{z}_i] + \frac{1}{2}e^{2\pi\mathbf{a}_i\mathbf{z}_i}H[-\mathbf{z}_i],$$

*where* $H[x]$ *is the Heaviside function. Because* $\mathbf{g}_i(\mathbf{z})$ *is an even function in* $k_l$*,* $l \neq i$*, its total inverse Fourier transform is real.*

*Therefore, the inverse Fourier transform of* $\dfrac{\mathbf{k}_i\hat{\mathbf{F}}_j(\mathbf{k})}{(2\pi\mathrm{i})\cdot\sum\limits_l \mathbf{k}_l^2}$ *is the convolution of two real functions.*

# 3   2-D Case as a Simple Intuitive Example of the Construction

For 2-dimensional systems, we have:

$$\mathbf{k}_1\hat{\mathbf{F}}_1(\mathbf{k}) + \mathbf{k}_2\hat{\mathbf{F}}_2(\mathbf{k}) = 0, \tag{S.10}$$

and hence Eq. (S.9) has a simple form:

$$
\begin{aligned}
\mathbf{Q}_{21}(\mathbf{z}_1, \mathbf{z}_2) &= -\mathbf{Q}_{12}(\mathbf{z}_1, \mathbf{z}_2) \\
&= p^s(\mathbf{z}_1, \mathbf{z}_2)^{-1} \int_{\mathbf{z}_2^0}^{\mathbf{z}_2} \mathbf{f}_1(\mathbf{z}_1, s) p^s(\mathbf{z}_1, s) \mathrm{d}s \\
&\quad - \int_{\mathbf{z}_2^0}^{\mathbf{z}_2} \frac{\partial}{\partial \mathbf{z}_1} \Big( \mathbf{D}_{11}(\mathbf{z}_1, s) p^s(\mathbf{z}_1, s) \Big) - \frac{\partial}{\partial s} \Big( \mathbf{D}_{12}(\mathbf{z}_1, s) p^s(\mathbf{z}_1, s) \Big) \mathrm{d}s \\
&= -p_s(\mathbf{z}_1, \mathbf{z}_2)^{-1} \int_{\mathbf{z}_1^0}^{\mathbf{z}_1} \mathbf{f}_2(s, \mathbf{z}_2) p^s(s, \mathbf{z}_2) \mathrm{d}s \\
&\quad + \int_{\mathbf{z}_1^0}^{\mathbf{z}_1} \frac{\partial}{\partial s} \Big( \mathbf{D}_{21}(s, \mathbf{z}_2) p^s(s, \mathbf{z}_2) \Big) + \frac{\partial}{\partial \mathbf{z}_2} \Big( \mathbf{D}_{22}(s, \mathbf{z}_2) p^s(s, \mathbf{z}_2) \Big) \mathrm{d}s.
\end{aligned}
\tag{S.11}
$$

## 4 Previous MCMC Algorithms in the Form of Continuous Markov Processes as Elements in the Current Recipe

This section parallels that of Sec. 3.1 of the main paper, but in terms of the continuous dynamics underlying the samplers. This allows us to rapidly draw connections with our SDE framework of Sec. 2.1. Fig. S.1 provides a cartoon visualization of the portion of the product space $\mathbf{D}(\mathbf{z}) \times \mathbf{Q}(\mathbf{z})$ already covered by past methods, after casting these methods in our framework below. Our proposed gSGRHMC method covers a portion of this space previously not explored.

Figure S.1: Cartoon of how previous methods explore the space of possible $\mathbf{D}(\mathbf{z})$ and $\mathbf{Q}(\mathbf{z})$ matrices, along with our proposed gSGRHMC method of Sec. 3.2.

**Hamiltonian Monte Carlo (HMC)** The continuous dynamics underlying Eq. (10) in the main paper are

$$
\begin{cases}
\mathrm{d}\theta = \mathbf{M}^{-1} r \mathrm{d}t \\
\mathrm{d}r = -\nabla U(\theta) \mathrm{d}t.
\end{cases}
\tag{S.12}
$$

Again, we see Eq. (S.12) is a special case of our proposed framework with $\mathbf{z} = (\theta, r)$, $H(\theta, r) = U(\theta) + \frac{1}{2} r^T M^{-1} r$, $\mathbf{Q}(\theta, r) = \begin{pmatrix} 0 & -I \\ I & 0 \end{pmatrix}$ and $\mathbf{D}(\theta, r) = \mathbf{0}$.

**Stochastic Gradient Hamiltonian Monte Carlo (SGHMC)** As described in [2], replacing $\nabla U(\theta)$ by the stochastic gradient $\nabla \widetilde{U}(\theta)$ in the $\epsilon$-discretized HMC system of Eq. (10) (resulting in Eq. (11)) has a continuous-time representation as:

$$
\text{Naive}: \begin{cases}
\mathrm{d}\theta = \mathbf{M}^{-1} r \mathrm{d}t \\
\mathrm{d}r = -\nabla U(\theta) \mathrm{d}t + \sqrt{\epsilon \mathbf{V}(\theta)} \mathrm{d}\mathbf{W}(t) \approx -\nabla \widetilde{U}(\theta) \mathrm{d}t.
\end{cases}
\tag{S.13}
$$

Analogously to Sec. 3.1, these dynamics do not fit into our framework. Instead, in our framework we see that the noise term $\sqrt{2\mathbf{D}(\mathbf{z})}\mathrm{d}\mathbf{W}(t)$ is paired with a $\mathbf{D}(\mathbf{z})\nabla H(\mathbf{z})$ term, hinting that such a term must be added to the dynamics of Eq. (S.13). Here, $\mathbf{D}(\theta, r) = \begin{pmatrix} 0 & 0 \\ 0 & \epsilon\mathbf{V} \end{pmatrix}$, which means we need to add a term of the form $\mathbf{D}(\mathbf{z})\nabla H(\mathbf{z}) = \epsilon\mathbf{V}\nabla_r H(\theta, r) = \epsilon\mathbf{V}\mathbf{M}^{-1}r$. Interestingly, this is the correction strategy proposed in [2], but through a physical interpretation of the dynamics. In particular, the term $\epsilon\mathbf{V}\mathbf{M}^{-1}r$ (or, generically, $\mathbf{C}\mathbf{M}^{-1}r$) has an interpretation as a friction term and leads to second order Langevin dynamics:

$$\begin{cases} \mathrm{d}\theta = \mathbf{M}^{-1}r\mathrm{d}t \\ \mathrm{d}r = -\nabla U(\theta)\mathrm{d}t - \mathbf{C}\mathbf{M}^{-1}r\mathrm{d}t + \sqrt{2\mathbf{C}-\epsilon\mathbf{V}(\theta)}\mathrm{d}\mathbf{W}(t) + \sqrt{\epsilon\mathbf{V}(\theta)}\mathrm{d}\mathbf{W}(t). \end{cases} \tag{S.14}$$

This method now fits into our framework with $H(\theta, r)$ and $\mathbf{Q}(\theta, r)$ as in HMC, but here with $\mathbf{D}(\theta, r) = \begin{pmatrix} 0 & 0 \\ 0 & C \end{pmatrix}$.

**Stochastic Gradient Langevin Dynamics (SGLD)**    SGLD [6] proposes to use the following first order (no momentum) Langevin dynamics to generate samples

$$\mathrm{d}\theta = -\mathbf{D}\nabla U(\theta)\mathrm{d}t + \sqrt{2\mathbf{D}}\,\mathrm{d}\mathbf{W}(t). \tag{S.15}$$

This algorithm corresponds to taking $\mathbf{z} = \theta$ with $H(\theta) = U(\theta)$, $\mathbf{D}(\theta) = \mathbf{D}$, $\mathbf{Q}(\theta) = 0$. As in the case of SGHMC, the variance of the stochastic gradient can be subtracted from the sampler injected noise $\sqrt{2\mathbf{D}}\mathbf{W}(t)$ to make the finite stepsize simulation more accurate. This variant of SGLD leads to the stochastic gradient Fisher scoring algorithm [1].

**Stochastic Gradient Riemannian Langevin Dynamics (SGRLD)**    SGLD can be generalized to use an adaptive diffusion matrix $\mathbf{D}(\theta)$. Specifically, it is interesting to take $\mathbf{D}(\theta) = \mathbf{G}^{-1}(\theta)$, where $\mathbf{G}(\theta)$ is the Fisher information metric. The sampler dynamics is given by

$$\mathrm{d}\theta = -\mathbf{G}^{-1}(\theta)\nabla U(\theta)\mathrm{d}t + \Gamma(\theta) + \sqrt{2\mathbf{G}^{-1}(\theta)}\mathrm{d}\mathbf{W}(t). \tag{S.16}$$

Taking $\mathbf{D}(\theta) = \mathbf{G}^{-1}(\theta)$ and $\mathbf{Q}(\theta) = \mathbf{0}$, the SGRLD method falls into our current framework with the correction term $\Gamma_i(\theta) = \sum_j \frac{\partial \mathbf{D}_{ij}(\theta)}{\partial \theta_j}$.

**Stochastic Gradient Nosé-Hoover Thermostat (SGNHT)**    Finally, the continuous dynamics underlying the SGNHT [3] algorithm in Sec. 3.1 are

$$\begin{cases} \mathrm{d}\theta = r\mathrm{d}t \\ \mathrm{d}r = -\nabla U(\theta)\mathrm{d}t - \xi r\,\mathrm{d}t + \sqrt{2A}\,\mathrm{d}\mathbf{W}(t) \\ \mathrm{d}\xi = \left(\frac{1}{d}r^T r - 1\right)r\mathrm{d}t. \end{cases} \tag{S.17}$$

Again, we see we can take $\mathbf{z} = (\theta, r, \xi)$, $H(\theta, r, \xi) = U(\theta) + \frac{1}{2}r^T r + \frac{1}{2d}(\xi - A)^2$, $\mathbf{D}(\theta, r, \xi) = \begin{pmatrix} 0 & 0 & 0 \\ 0 & A \cdot \mathbf{I} & 0 \\ 0 & 0 & 0 \end{pmatrix}$, and $\mathbf{Q}(\theta, r, \xi) = \begin{pmatrix} 0 & -\mathbf{I} & 0 \\ \mathbf{I} & 0 & r/d \\ 0 & -r^T/d & 0 \end{pmatrix}$ to place these dynamics within our framework.

## 5   Discussion of Choice of D and Q

A lot of choices of $\mathbf{D}(\mathbf{z})$ and $\mathbf{Q}(\mathbf{z})$ could potentially result in faster convergence of the samplers than those previously explored. For example, $\mathbf{D}(\mathbf{z})$ determines how much noise is introduced. Hence, an adaptive diffusion matrix $\mathbf{D}(\mathbf{z})$ can facilitate a faster escape from a local mode if $||\mathbf{D}(\mathbf{z})||$ is larger in regions of low probability, and can increase accuracy near the global mode if $||\mathbf{D}(\mathbf{z})||$ is smaller in regions of high probability. Motivated by the fact that a majority of the parameter space is covered by low probability mass regions where less accuracy is often needed, one might want to traverse these regions quickly. As such, an adaptive curl matrix $\mathbf{Q}(\mathbf{z})$ with 2-norm growing with the level set of the distribution can facilitate a more efficient sampler. We explore an example of this in the gSGRHMC algorithm of the synthetic experiments (see Supp. 6.1).

# 6 Parameter Settings in Synthetic and Online Latent Dirichlet Allocation Experiments

## 6.1 Synthetic Experiments

In the synthetic experiment using gSGRHMC, we specifically consider $\mathbf{G}(\theta)^{-1} = D\sqrt{|\widetilde{U}(\theta) + C|}$. The constant $C$ ensures that $\widetilde{U}(\theta) + C$ is positive in most cases so that the fluctuation is indeed smaller when the probability density function is higher. Note that we define $\mathbf{G}(\theta)$ in terms of $\widetilde{U}(\theta)$ to avoid a costly full-data computation. We choose $D = 1.5$ and $C = 0.5$ in the experiments. The design of $\mathbf{G}$ is motivated by the discussion in Supp. 5, taking $\mathbf{Q}(\theta)$ to have 2-norm growing with the level sets of the potential function can lead to faster exploration of the posterior.

Figure S.2: For two simulated 1D distributions (`black`) defined by $U(\theta) = \theta^2/2$ (*left*) and $U(\theta) = \theta^4 - 2\theta^2$ (*right*), comparison of `SGLD`, `SGHMC`, the `naïve SGRHMC` of Eq. (16), and the `gSGRHMC` of Eq. (17) in the main paper.

Comparison of SGLD, SGHMC, the naïve implementation of SGRHMC (Eq. (16)), and the gSGRHMC methods is shown in Fig. S.2, indicating the incorrectness of the naïve SGRHMC.

## 6.2 Online Latent Dirichlet Allocation Experiment

In the online latent Dirichlet allocation (LDA) experiment, we used minibatches of 50 documents and $K = 50$ topics. Similar to [4], the stochastic gradient of the log posterior of the parameter $\theta$ on a minibatch $\widetilde{\mathcal{S}}$ is calculated as

$$\frac{\partial \log p(\theta|\mathbf{x}, \alpha, \gamma)}{\partial \theta_{kw}} \approx \frac{\alpha - 1}{\theta_{kw}} - 1 + \frac{|\mathcal{S}|}{|\widetilde{\mathcal{S}}|} \sum_{d \in \widetilde{\mathcal{S}}} \mathbb{E}_{\mathbf{z}^{(d)}|\mathbf{x}^{(d)}, \theta, \gamma} \left[ \frac{n_{dkw}}{\theta_{kw}} - \frac{n_{dk\cdot}}{\theta_{k\cdot}} \right], \qquad (S.18)$$

where $\alpha$ is the hyper-parameter for the Gamma prior of per-topic word distributions, and $\gamma$ for the per-document topic distributions. Here, $n_{dkw}$ is the count of how many times word $w$ is assigned to topic $k$ in document $d$ (via $z_j^{(d)} = k$ for $x_j = w$). The $\cdot$ notation indicates $n_{dk\cdot} = \sum_w n_{dkw}$. To calculate the expectation of the latent topic assignment counts $n_{dkw}$, Gibbs sampling is used on the topic assignments in each document separately, using the conditional distributions

$$p(z_j^{(d)} = k|\mathbf{x}^{(d)}, \theta, \gamma) = \frac{\left(\gamma + n_{dk\cdot}^{\backslash j}\right) \theta_{kx_j^{(d)}}}{\sum_k \left(\gamma + n_{dk\cdot}^{\backslash j}\right) \theta_{kx_j^{(d)}}}, \qquad (S.19)$$

where $\backslash j$ represents a count excluding the topic assignment variable $z_j^{(d)}$ being updated. See [4] for further details.

We follow the experimental settings in [4] for Riemmanian samplers (SGRLD and SGRHMC), taking the hyper-parameters of Dirichlet priors to be $\gamma = 0.01$ and $\alpha = 0.0001$. Since the non-Riemmanian samplers (SGLD and SGHMC) do not handle distributions with mass concentrated over small regions as well as the Riemmanian samplers, we found $\gamma = 0.1$ and $\alpha = 0.01$ to be optimal hyper-parameters for them and use these instead for SGLD and SGHMC. In doing so,

we are modifying the posterior being sampled, but wished to provide as good of performance as possible for these baseline methods for a fair comparison. For the SGRLD method, we keep the stepsize schedule of $\epsilon_t = \left( a \cdot \left( 1 + \frac{t}{b} \right) \right)^{-c}$ and corresponding optimal parameters $a, b, c$ used in the experiment of [4]. For the other methods, we use a constant stepsize because it was easier to tune. (A constant stepsize for SGRLD performed worse than the schedule described above, so again we are trying to be as fair to baseline methods as possible when using non-constant stepsize for SGRLD.) A grid search is performed to find $\epsilon_t = 0.02$ for the SGRHMC method; $\epsilon_t = 0.01$, $\mathbf{D} = I$ (corresponding to Eq. (13) in the main paper) for the SGLD method; and $\epsilon_t = 0.1$, $\mathbf{C} = \mathbf{M} = I$ (corresponding to Eq. (12) in the main paper) for the SGHMC method.

For a randomly selected subset of topics, in Table S.1 we show the top seven most heavily weighted words in the topic learned with the SGRHMC sampler.

| | | | | | | |
|---|---|---|---|---|---|---|
| "ENGINES" | speed | product | introduced | designs | fuel | quality |
| "ROYAL" | britain | queen | sir | earl | died | house |
| "ARMY" | commander | forces | war | general | military | colonel |
| "STUDY" | analysis | space | program | user | research | developed |
| "PARTY" | act | office | judge | justice | legal | vote |
| "DESIGN" | size | glass | device | memory | engine | cost |
| "PUBLIC" | report | health | community | industry | conference | congress |
| "CHURCH" | prayers | communion | religious | faith | historical | doctrine |
| "COMPANY" | design | production | produced | management | market | primary |
| "PRESIDENT" | national | minister | trial | states | policy | council |
| "SCORE" | goals | team | club | league | clubs | years |

Table S.1: The top seven most heavily weighted words (columns) associated with each of a randomly selected set of 11 topics (rows) learned with the SGRHMC sampler from 10,000 documents (about 0.3% of the articles in Wikipedia). The capitalized words in the first column represent the most heavily weighted word in each topic, and are used as the topic labels.