[Reviews · NeurIPS 2015]

Submitted by Assigned_Reviewer_1

The authors propose a general framework for designing new MCMC samplers, including methods that use stochastic gradients. Their approach is to define a stochastic dynamical system whose stationary distribution is the target distribution from which we want to sample from. The stochastic dynamical system is represented through a stochastic differential equation that is simulated through an epsilon-discretization approach. As the step-size parameter epsilon goes to zero, the bias in the simulation vanishes. The proposed approach can handle stochastic approximations obtained by sub-sampling the data in mini-batches. For this, the additional noise produced by the sub-sampling process is removed from the noise used to simulate the SDE. The proposed framework constructs the SDE using two state dependent matrices D(z) and Q(z) and by modifying these matrices the authors can explore the space of MCMC methods. The authors proof that their approach samples from the correct stationary distribution as the step size epsilon tends to zero. The authors also proof that any continuous Markov process with a particular stationary distribution can be represented in their framework and consequently, such framework is complete. After this, the authors show how several already existing samplers can be represented in their framework: Hamiltonian Monte Carlo, Stochastic Gradient Hamiltonian Monte Carlo, Stochastic Gradient Langevin Dynamics, Stochastic Gradient Riemannian Langevin Dynamics and Stochastic Nose-Hoover Thermostast. Finally, the authors proposed based on their framework a new sampler technique called Stochastic Gradient Riemann Hamiltonian Monte Carlo (SGRHMC). This method uses the Ficsher information matrix to precondition the dynamics through the matrices D(z) and Q(z). The SGRHMC method is shown to provide very good results on experiments with simulated data and streeming data extracted from Wikipedia.

Quality:

The paper is technically very sound, with theorems showing the applicability of the framework to any target distribution and its completeness in terms of how many continuous Markov processes it can represent.

Clarity:

The paper is clearly written and well organized. A few things could be improved for clarity: How do you construct in practice the matrix \hat{B}_t? When you substract \hat{B}_t to eliminate the stochastic gradient noise term, what happens if you get a covariance matrix that is not possitive definite? Perhaps this is not possible if \epsilon is small. In line 232 the minus sign in Q should probably be changed from the upper right identity matrix to the lower left one.

Originality:

The proposed framework is original. It allows the authors to re-design existing MCMC samples and propose new ones with better properties. The author's contribution is significantly different from previous work.

Significance:

This work is highly significant, specifying a principled way to design new scalable MCMC samples that work by sub-sampling the data and are able to sample from the correct target distribution in the limit of the step size tending to zero.
Summary: This is a strong paper, proposing a general framework for designing scalable and correct MCMC samplers. The proposed framework comes with theoretical guarantees and the authors illustrate its usefulness in experiments with simulated and real-world data.

Submitted by Assigned_Reviewer_2

The general theme of the paper is the development of a general recipe for constructing MCMC samplers motivated by Markov processes while avoiding a Metropolis-Hastings correction step. The recipe allows the recovery of many existing stochastic gradient MCMC algorithm and

the authors use it to propose a new

algorithm.

Comments:

Consider Theorem 2.

A phrase like "(a condition usually satisfied)" does not belong in the statement of a theorem.

Also, integrable with respect to which measure?

I'm not sure I see the point of Figure 1.

All of this information was conveyed clearly in the paper.

In line 203 on page 4, the ordering has not been defined as far as I can tell.

My only real criticism of the paper is that the the examples in

section 4 are given a somewhat cursory treatment.

It would seem to be difficult to reproduce the results here with the information conveyed.

Also, there is no discussion of the computational effort required for any of the methods.

Typos:

Line 235 page 5:

"will results" -> "will result"

Line 246 page 5: "need add" -> "need to add"
Summary: This is a well-written paper on an interesting, significant topic.

My main criticism of the paper is that the the examples in

section 4 are given a somewhat cursory treatment.

However, it would seem to require only a minor revision.

Submitted by Assigned_Reviewer_3

Summary: This paper proposes a complete framework that summarizes several old methods. Afterwards, it demonstrates an example of concretely applying this framework to practice.

Quality: The theoretical part of the paper is persuasive and sheds light on potential implication of this work as a pivotal summary of the field. Given its immature state of real application mentioned in its introduction, it is probably more appropriate to be viewed as a theoretical summary. Admittedly, it is crucial to have this summary. The domain knowledge remains needed for this framework to be useful, which is not quite different from those old methods. The experiment is helpful with demonstrating its idea, but the importance of this work cannot be revealed by those experiments. Indeed, it is not trivial to design relevant experiments given an incomplete but potentially powerful algorithm.

Clarity: The presentation of the work is good. Originality: It is a nice piece of work as a summary with fair amount of work to fill gaps from different methods. Significance: Its importance is undoubtable, and it should have potential audience.

Summary: This paper emphasizes a complete theoretical framework that can naturally encompass several old methods. While its theoretical derivation is solid and its potential implication in practical applications is possible, the missing piece mentioned in its introduction makes it difficult to apply, comparing with other simpler methods, thus being amere theoretical summary.

Submitted by Assigned_Reviewer_4

The authors motivate the paper with a well referenced introduction where they review most of the main stochastic gradient MCMC methods. They define two matrices, D and Q, that they will particularize for different existing methods. The paper provides two theorems showing that 1) a pair of equations that yield the Markov process to the correct stationary distribution 2) a proof that any continuos Markov process with the correct stationary distribution necessarily fulfill the proposed system of equations (completeness of the framework). The authors situate known and very relevant algorithms within the framework, providing the matrices D and Q of each method (e.g., HMC and SGLD). A new algorithm, SGRHMC, is proposed and its performance is analyze in Section 4.

- Quality: The work seems to be technically solid, although I do not completely follow both proofs. - Clarity: The paper is well written, and its clarity is a strong point of the paper. The introduction is well grounded and the purpose and the contribution are clear at any time. - Originality: This is the weaker point of the work, since it "rediscovers" existing algorithms. I am not sure about the results of the proposed algorithm. The results section is very short and not deeply explained, probably due to space problems.

- Significance: I totally support works like this, where many known methods are explained in a unified way, allowing the community to understand them better and have new insights. I think this paper is very interesting, even if it did not propose any new algorithm.
Summary: The paper established a unified framework for stochastic gradient MCMC methods, reinventing previous methods and proposing a new one. The paper is well written and interesting, and therefore, I recommend its acceptance.

Author Feedback
Author rebuttal: We thank the reviewers for their thoughtful and positive feedback. In a revision, we will incorporate the helpful suggestions made, especially to provide more details on the experiments.

We clarify upfront that our primary contributions go beyond a theoretical framework for unifying past (and future) methods. In addition to this (nontrivial) contribution, our framework enables quickly devising new (and correct) samplers. We demonstrate this with our proposed SGRHMC algorithm, which was a sought-after sampler in the community that nobody had been able to correctly specify. We show that this new sampler outperforms previous samplers in a streaming LDA analysis. This is a very real and important application; perhaps we were too understated in this section. We will make these points clearer.

R1:
- How to construct \hat{B}?

\hat{B} can be calculated using an empirical Fisher information estimator, such as runtime moment estimation. It can also be set to 0 due to the small stepsize setting.

- When you subtract \hat{B}_t ..., what happens if you get a covariance matrix that is not positive definite?

Indeed, with small enough eps we can always get a PSD result. Additionally, the stochastic gradient noise scales down faster than the added noise.

R2:
- Theorem 2 suggestions

We will move the parenthetical statement outside the theorem and state that we integrate wrt the Lebesgue measure.

- I'm not sure I see the point of Fig 1. All of this information was conveyed clearly in the paper.

We find Fig. 1(left) helpful in guiding the text. To save space, we will move Fig. 1(right) to the Supplement and instead expand the experiments section text.

- In line 203 on page 4, the ordering has not been defined

A >= B indicates A-B is PSD, which we will clarify in the text.

- My only real criticism of the paper is that the the examples in section 4 are given a somewhat cursory treatment. It would seem to be difficult to reproduce the results here... Also, there is no discussion of the computational effort required for any of the methods.

We agree that we should make a better space tradeoff in a revision. We reported parameter settings in the synthetic experiments. Although we mention following the settings of [12] for the online LDA experiment, these settings need to be explicitly stated. We will also provide runtimes and a computational complexity analysis of the methods in the Supplement, and make our source code publicly available.

R3:
- Originality: This is the weaker point of the work, since it "rediscovers" existing algorithms. I am not sure about the results of the proposed algorithm. The results section is very short and not deeply explained, probably due to space problems.

We appreciate your support of this work. To clarify its originality, we see this as going beyond "rediscovering" existing algorithms. In addition to our unifying framework, our SGRHMC algorithm is in and of itself a novel and important contribution. The existence of such an algorithm has been hotly discussed. We apologize if the importance of our experimental results was not clear enough; we will expand this text. Likewise, our framework is a practical tool for straightforwardly devising and testing out other new samplers.

R4:
- While its theoretical derivation is solid and its potential implication in practical applications is possible, the missing piece mentioned in its introduction makes it difficult to apply..., thus being a mere theoretical summary.

Indeed, we do not provide a method to automatically search over all possible D(z), Q(z) (an unboundedly large and challengingly structured space). Instead, we propose using our derived D(z),Q(z) building blocks (or intuitive variants thereof) to devise new samplers (as in our proposed SGRHMC). This is akin to the automatic model selection challenge in machine learning, which is completely unsolved: Instead of searching over all possible models, modelers use existing building blocks to devise new models. We do not view this as a fundamental shortfall of our contribution, but rather an interesting and challenging direction for future research.

We emphasize that our paper is more than a theoretical summary. Eq.(9) provides a practical algorithm for deploying samplers with new D(z),Q(z); we demonstrate this with our novel SGRHMC in Secs. 3.3 and 4.

R6:
- Interesting paper ... I am not an expert in this particular area but it feels to me there have been backed with more examples.

We appreciate the positive feedback, and understand the desire to have more examples. The focus of this paper is on deriving a general framework that can be used to design new SGMCMC algorithms. The experiments justify the correctness of the approach, as well as the effectiveness of the framework in quickly devising new algorithms (such as our SGRHMC) that outperform previous samplers in important, real-world applications (e.g., streaming LDA experiments).